# Scale Measurement of Health Primary Service Utilization among the Migrant International Population

Consuelo Cruz-Riveros [1,*], Alfonso Urzúa [1] and Carolina Lagos [2]

1    Escuela de Psicología, Universidad Católica del Norte, Antofagasta 1270709, Chile
2    Escuela de Enfermería, Universidad Santo Tomás, Sede Santiago, Santiago 8370003, Chile
*    Correspondence: consuelocruzri@santotomas.cl; Tel.: +56-966797759

**Abstract:** In this article, we analyze the internal structure of the scale for experience in exercising the right to health care (EERHC), based on the focus from the World Health Organization (WHO) on the right to health care, from the perspective of international migrants, in Chile. The methodology was an instrumental study ($n$ = 563) conducted to analyze the psychometric properties of the EERHC scale. Its reliability and internal consistency were evaluated, while the exploratory structural equation modeling (ESEM) model and confirmatory factor analysis (CFA) were used to identify the structure of relationships between the variables measured. The item–dimension correlations obtained present levels of r $\geq$ 0.3, and the Cronbach's $\alpha$ and McDonald's $\omega$ presented ranges >0.9, considered to be acceptable on all models. Results: the model was selected for presenting a good fit index $\chi^2$ = 24,850, df = 300, $p$ = 0.000; RMSEA = 0.07; CFI = 0.97; TLI = 0.95; and SRMR = 0.03. The evidence obtained lets us conclude that the scale has forty-five items and four dimensions. The findings demonstrate a good internal structure and are useful to measure primary health care service utilization based on the framework.

**Keywords:** transients and migrants; health care; health personnel; primary health care

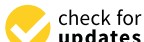



## 1. Introduction

International migration is defined as the movement of any person between states by crossing its borders [1]. This movement has led to around 281 million people residing in a different country from where they were born, which is 3.6% of the world population [1].

There is abundant evidence about the relation between migration and health, drawn from both researchers on the ground and global-level health organizations, particularly the World Health Organization (WHO) [2–6]. The WHO has established migration as a social health determinant, due to the association it can have with health maintenance, improvement, or deterioration in the short or long term [3,5,6]. Some of the factors reported as causes are associated with difficulties faced during movement (migration process) which may depend on the way they enter a country (people who enter illegally may suffer various types of violence during their migration: sexual, robberies, and abuses) [7,8]. After this, once they reside in a new country, they must adapt, and often suffer lifestyle changes [4–6]. Regardless of their health status at their moment of entry, one of their fundamental rights is health care, meaning that they should have access to use various health service systems and services in their new resident country, across various care levels [6].

Health care service use is considered to be a process involving various interventions, facilitating responses to human health needs [9,10]. According to current global rules, health care as a fundamental right must allow all people the possibility to achieve the greatest possible enjoyment of their mental and physical health status, within a process considering due respect for the principles of equality and non-discrimination [6,9–11].

The strategies established by the Committee on Economic, Social and Cultural Rights of the WHO, in their General Observation #14, indicate the consideration of four dimensions to

fulfill the right to health care linked with health service utilization: availability, accessibility, acceptability, and quality and safety [5,10]. Availability establishes a guarantee of general supplies such as establishments, goods and public health services [10]. Existing barriers include historic economic deficits impeding provision of supplies, equipment, proper care infrastructure, problems with hiring personnel as cultural facilitators, and low care hour availability [9,12–15].

Accessibility involves the principle of non-discrimination in health practices within various dimensions including access to health centers, goods and services, safe physical accessibility for the population (throughout their life cycle and in people with any physical or cognitive difficulties), access to information and economic accessibility (which guarantees health care for all people) [10]. The most reported barriers are perceptions of discrimination, prejudices and beliefs among personnel, lack of access to information about health care system functions, fees and charges, and health care sites' safety [9,11–20].

Acceptability implies that the establishments, goods and services must use practices which consider a gender focus, an intercultural focus (language and culture alike) and medical ethics [10]. Some factors studied include using jargon, language, user treatment, and cultural care adjustment [19–21].

Finally, the quality and safety dimension includes adequate services and supplies from a medical and scientific perspective, and which are also of good quality, considering professionals trained at all levels as well [10]. In this dimension, some of the indicators used include user satisfaction, and accreditation indicators focused on quality and safety [11–13].

In this context, making the right to health care concrete can be carried out through service in health care, whose evaluation is currently carried out through various indicators established by different systems and health studies. These include coverage for services, and quality and safety indicators, which tend to be measured in an isolated way [11,13]. Some instruments used for these isolated measures are SERVQUAL (Service Quality), which measures user satisfaction, PCAS (Primary Care Assessment Survey), which evaluates strengths and weaknesses of primary health care, the Care Questionnaire, which evaluates continuity of care, or the CICAA scale, which evaluates care centered on the use and the effective access model [16–18,22–25]. This final point, along with the utilization model from Anderson and Aday, has tried to integrate the human social context, although there are some elements which are still not considered that are still part of the right to health care [16–18].

Some reported acceptability indicators are associated with practices implemented in health care establishments which include intercultural facilitators, language adaptation and jargon-free language, which act as service usage facilitators [16,17,17,18,18–26]. However, there are also factors that act as barriers to service use, such as lack of knowledge about interculturality, regulation implementation, care that does not meet the needs, dissatisfaction with services, patients' beliefs and perspectives by health team members, discrimination associated with migratory status and/or economic aspects [9,14–18,22–27].

Considering the previous point, even when observing that there are different instruments which can gather diverse elements tied with exercising the right to health care, extant measurements consider the evaluation of this right from the fragmentation of the indicator, that is, by covering only some of the components suggested by the WHO [28–30].

Given the need for an integral evaluation of the various dimensions, the aim of this present article is to analyze the internal structure of the EERHC scale, based on the focus from the WHO on the right to health care, from the perspective of international migrants, in Chile. This scale will allow us to identify and explain the factors tied with primary health care service use from the four dimensions established by the right to health care.

## 2. Materials and Methods

### 2.1. Design

The present research is an instrumental study, a design with general directives guaranteeing the fulfillment of minimal scientific properties for instrument design/adaptation [25]. This design type has internationally accepted regulations for instrument construction [25].

### 2.2. Participants

The sample is non-probabilistic and was composed of 563 international migrants, who were recruited between January and July 2022 in the Metropolitan, Antofagasta and Biobío Regions in Chile. The inclusion criteria were being at least 18 years old, speaking Spanish, and having used a primary health care service. The exclusion criteria were the presence of cognitive health problems such as dementia and other pathologies impeding comprehension of the survey being applied. The median age was 37.5 years (DT = 12.4), 327 (58.1%) were women, 313 (55.6%) single individuals, 228 (40.5%) people were from Venezuela, and 136 (24,2%) were from Colombia. The education level for 276 (49%) people was secondary school, and 431 (76.6%) people had normalized migratory status.

### 2.3. Procedures

The Project was approved by the Scientific Ethics Committee of Universidad Católica del Norte (resolution 015/2021). All participants signed voluntary informed consent for the study.

Five phases were carried out in the study:

Phase 1 Systematic review about factors related with exercising the right to health care in service use [31].

Phase 2 Conceptual definition, considered the process of developing the operative, semantic and syntactic definition of the service utilization variable, from the perspective of health rights, which defined both the central concept and the dimensions which must be included in the instrument [31]. Five dimensions were established with their corresponding subcategories (Table 1).

**Table 1.** Dimension definitions and total items, from the EERHC instrument.

| Dimensions | Total Items |
|---|---|
| Availability: understood as the "guarantee of general supplies, including establishments, goods and services" [10]. | 17 |
| Accessibility: considered "the principle of non-discrimination, in the use of services including establishments, goods and portfolio of services, governance, geographical and temporal, which does not jeopardize the population regardless of age, gender and legal status in the country, for the provision of prevention, curing, rehabilitation and promotion services" [10]. | 13 |
| Acceptability: defined as "actions focused on gender, cultural adaptation and respect for medical ethics, also including actions which help decrease language and cultural barriers, including cultural mediation, interpretation and translation" [10]. | 15 |
| Quality: "ensuring the minimum standard established, allowing for safe care for people within environments covered by availability" [10]. | 11 |

Phase 3 Construction of the items—since several were measured in a segmented way in accordance with recommendations from Muñiz et al. (2005), we sought to have: representativeness, relevance, diversity, clarity, simplicity and comprehensibility. A total of 56 reagents were designed for the instrument, aimed at gathering perceptions from international migrants. Based on the conceptual definition, these reagents were organized in the established dimensions [31].

Phase 4 Initial evaluation, of a qualitative nature, which consisted, on the one hand, of cognitive interviews among the target population, in order to evaluate the drafting and

comprehension of each preliminary instrument item [31] and, on the other, of the review by three researchers who were experts in the migrant health field. The changes suggested by participants regarding reagent composition were carried out.

Phase 5 Application, where the field phase was carried out with the target population, considering the recommendations of a minimum of 5 people per item [31].

### 2.4. Instrument

The scale for experience in exercising the right to health care [EERHC] is composed of 56 reagents grouped into the four dimensions of availability, accessibility, acceptability, and quality and safety (Table 1). Each reagent is evaluated with a score from 1 to 4 (1 "completely disagree", 2 "disagree", 3 "agree" and 4 "completely agree").

### 2.5. Ethical Considerations

The study was approved by the ethics committee of Universidad Católica del Norte. It was governed by the principles of voluntariness, confidentiality, and participant anonymity, which was reflected through the signing of informed consent.

### 2.6. Data Analysis

Descriptive statistics (median, standard deviation, asymmetry and kurtosis) were estimated via the SPSS24 program. The JAMOVI program was used for reliabilities via the McDonald's ω coefficient, Cronbach's α and item–dimension correlation. The exploratory structural equation modeling (ESEM) is used to determine the factorial structure of a scale. ESEM combines exploratory factor analysis (EFA) with confirmatory factor analysis (CFA) and uses the variance–covariance matrices that best fit the type of scale in question. Additionally, this method does not require the factor loading of items on other factors to be zero, allowing for a more accurate calculation of fit indices and correlations between latent variables [32]. Regarding the rotation method used, target was chosen. It is a factor analysis technique that aims to obtain a factor structure that fits a specific correlation matrix. This technique is used when there is a clear idea of the correlations that should exist between the factors. It differs from traditional oblique and orthogonal rotation techniques in that they seek a factor structure that maximizes the explained variance or minimizes the correlation between factors. In contrast, target rotation focuses on adjusting the factor structure to a given correlation matrix [32].

Later, the factorial structure of the scale was analyzed using confirmatory factorial analysis. Estimation of fit parameters used a factorial load analysis procedure carried out with Mplus, using the weighted least squares method with the median estimation method (WLSMV) due to the ordinal nature of the scale and due to being a robust estimator which does not assume a normal distribution [33]. The model fit was interpreted according to the fit indices with cutoff points of CFI > 0.90; TLI > 0.90; RMSEA between 0.05 and 0.08 and SRMR < 0.08 [31,34,35].

## 3. Result

### 3.1. Preliminary Analysis of Items

Table 2 shows the descriptive statistics of the 56 items from the initial EERHC scale. We can observe that the item acep14 has the highest average score (M = 3.39), while item disp14 has the lowest (M = 2.63). Regarding variability, item ac12 shows the greatest dispersion (DE = 0.92). When considering the distribution of answers from international migrants, asymmetry values for each item were negative overall, and kurtosis was negative overall. The asymmetry and kurtosis values considered acceptable for each item must not be any greater than the range > ±1.5 [35]. The asymmetry values thus lie within the ranges, while in kurtosis, the items that fall outside the established range are acep10 and acep14. The item–dimension correlations obtained present levels of r ≥ 0.3, considered appropriate for discriminating [35]. Another element used as a discrimination criterion was global internal

consistency according to dimensions. In both cases, the Cronbach's α and McDonald's ω presented ranges >0.9, considered to be acceptable [36].

**Table 2.** Preliminary analysis of EERHC scale items.

| Items | M | D.E | A | K | Correlation Item–Test | Cronbach's α Item–Test | McDonald's ω Item–Test | Cronbach's α Dimension | McDonald's ω Dimension |
|---|---|---|---|---|---|---|---|---|---|
| Disp1 | 3.3 | 0.72 | −0.99 | 1.11 | 0.62 | 0.97 | 0.97 | | |
| Disp2 | 3.31 | 0.72 | −0.91 | 0.71 | 0.58 | 0.97 | 0.97 | | |
| Disp3 | 3.11 | 0.82 | −0.67 | −0.09 | 0.59 | 0.97 | 0.97 | | |
| Disp4 | 3.19 | 0.79 | −0.74 | 0.06 | 0.67 | 0.97 | 0.97 | | |
| Disp5 | 3.16 | 0.82 | −0.68 | −0.2 | 0.61 | 0.97 | 0.97 | | |
| Disp6 | 3.29 | 0.76 | −0.91 | 0.43 | 0.61 | 0.97 | 0.97 | | |
| Disp7 | 3.26 | 0.77 | −0.9 | 0.5 | 0.67 | 0.97 | 0.97 | | |
| Disp8 | 3.14 | 0.82 | −0.83 | 0.28 | 0.67 | 0.97 | 0.97 | | |
| Disp9 | 3.14 | 0.82 | −0.73 | 0.14 | 0.67 | 0.97 | 0.97 | 0.94 | 0.94 |
| Disp10 | 3.05 | 0.89 | −0.73 | −0.03 | 0.61 | 0.97 | 0.97 | | |
| Disp11 | 2.96 | 0.85 | −0.53 | −0.44 | 0.65 | 0.97 | 0.97 | | |
| Disp12 | 3.0 | 0.91 | −0.58 | −0.26 | 0.68 | 0.97 | 0.97 | | |
| Disp13 | 2.87 | 0.89 | −0.46 | −0.57 | 0.67 | 0.97 | 0.97 | | |
| Disp14 | 2.86 | 0.86 | −0.45 | −0.53 | 0.6 | 0.97 | 0.97 | | |
| Disp15 | 3.05 | 0.8 | −0.67 | −0.17 | 0.66 | 0.97 | 0.97 | | |
| Disp16 | 3.1 | 0.9 | −0.69 | 0.09 | 0.73 | 0.97 | 0.97 | | |
| Disp17 | 2.95 | 0.76 | −0.54 | −0.49 | 0.6 | 0.97 | 0.97 | | |
| Ac1 | 3.08 | 0.76 | −0.63 | 0.23 | 0.47 | 0.97 | 0.97 | | |
| Ac2 | 3.1 | 0.81 | −0.62 | 0.18 | 0.46 | 0.97 | 0.97 | | |
| Ac3 | 3.1 | 0.8 | −0.82 | 0.4 | 0.51 | 0.97 | 0.97 | | |
| Ac4 | 3.0 | 0.88 | −0.63 | 0.13 | 0.38 | 0.97 | 0.97 | | |
| Ac5 | 2.98 | 0.83 | −0.62 | −0.28 | 0.49 | 0.97 | 0.97 | | |
| Ac6 | 3.1 | 0.76 | −0.75 | 0.1 | 0.55 | 0.97 | 0.97 | | |
| Ac7 | 3.31 | 0.76 | −1.1 | 1.15 | 0.64 | 0.97 | 0.97 | 0.9 | 0.9 |
| Ac8 | 3.3 | 0.79 | −1.04 | 0.94 | 0.68 | 0.97 | 0.97 | | |
| Ac9 | 3.29 | 0.76 | −1.13 | 1.05 | 0.69 | 0.97 | 0.97 | | |
| Ac10 | 3.28 | 0.82 | −1.02 | 0.9 | 0.68 | 0.97 | 0.97 | | |
| Ac11 | 3.25 | 0.76 | −1.1 | 0.88 | 0.63 | 0.97 | 0.97 | | |
| Ac12 | 3.25 | 0.92 | −0.8 | 0.22 | 0.66 | 0.97 | 0.97 | | |
| Ac13 | 2.87 | 0.81 | −0.52 | −0.5 | 0.48 | 0.97 | 0.97 | | |
| Acep1 | 2.96 | 0.81 | −0.46 | −0.27 | 0.59 | 0.97 | 0.97 | | |
| Acep2 | 2.95 | 0.81 | −0.5 | −0.14 | 0.64 | 0.97 | 0.97 | | |
| Acep3 | 2.95 | 0.82 | −0.47 | −0.2 | 0.68 | 0.97 | 0.97 | | |
| Acep4 | 3.25 | 0.77 | −0.95 | 0.3 | 0.60 | 0.97 | 0.97 | | |
| Acep5 | 3.26 | 0.75 | −0.86 | 0.31 | 0.68 | 0.97 | 0.97 | | |
| Acep6 | 3.35 | 0.75 | −1.09 | 0.97 | 0.66 | 0.97 | 0.97 | | |
| Acep7 | 3.32 | 0.77 | −1.1 | 1.15 | 0.66 | 0.97 | 0.97 | | |
| Acep8 | 3.21 | 0.73 | −0.91 | 0.67 | 0.67 | 0.97 | 0.97 | 0.94 | 0.95 |
| Acep9 | 3.31 | 0.7 | −1.07 | 1.25 | 0.72 | 0.97 | 0.97 | | |
| Acep10 | 3.37 | 0.71 | −1.16 | 1.63 | 0.71 | 0.97 | 0.97 | | |
| Acep11 | 3.38 | 0.71 | −1.1 | 1.3 | 0.72 | 0.97 | 0.97 | | |
| Acep12 | 3.3 | 0.68 | −0.92 | 0.9 | 0.71 | 0.97 | 0.97 | | |
| Acep13 | 3.34 | 0.69 | −0.93 | 1.06 | 0.72 | 0.97 | 0.97 | | |
| Acep14 | 3.39 | 0.69 | −1.2 | 1.97 | 0.69 | 0.97 | 0.97 | | |
| Acep15 | 3.36 | 0.72 | −1.0 | 1.13 | 0.68 | 0.97 | 0.97 | | |
| Cal1 | 3.27 | 0.73 | −0.94 | 1.09 | 0.72 | 0.97 | 0.97 | | |
| Cal2 | 3.25 | 0.86 | −1.0 | 1.26 | 0.54 | 0.97 | 0.97 | | |
| Cal3 | 3.09 | 0.74 | −0.81 | 0.1 | 0.71 | 0.97 | 0.97 | | |
| Cal4 | 3.27 | 0.74 | −0.97 | 1.0 | 0.69 | 0.97 | 0.97 | | |
| Cal5 | 3.28 | 0.74 | −0.93 | 0.74 | 0.7 | 0.97 | 0.97 | | |
| Cal6 | 3.25 | 0.73 | −0.88 | 0.76 | 0.7 | 0.97 | 0.97 | 0.93 | 0.94 |
| Cal7 | 3.28 | 0.72 | −0.92 | 0.76 | 0.71 | 0.97 | 0.97 | | |

**Table 2.** *Cont.*

| Items | M | D.E | A | K | Correlation Item–Test | Cronbach's α Item–Test | McDonald's ω Item–Test | Cronbach's α Dimension | McDonald's ω Dimension |
|---|---|---|---|---|---|---|---|---|---|
| Cal8 | 3.27 | 0.8 | −0.7 | 0.09 | 0.7 | 0.97 | 0.97 | | |
| Cal9 | 3.15 | 0.7 | −0.7 | −0.02 | 0.62 | 0.97 | 0.97 | | |
| Cal10 | 3.32 | 0.71 | −0.88 | 0.75 | 0.75 | 0.97 | 0.97 | | |
| Cal11 | 3.33 | 0.71 | −1.00 | 1.13 | 0.74 | 0.97 | 0.97 | | |

### 3.2. Structural Equation Model Exploratory

The initial analysis of the 54 items with exploratory structural equation model indicates inadequate fit ($\chi^2$ = 54,334, df = 1431, $p$ = 0.000; RMSEA= 0.09; CFI = 0.89; TLI = 0.88; and SRMR = 0.06). The deleted items presented standardized factor loadings <0.3; disp1, disp2, disp4, disp6, disp11, disp13, disp14, disp15, disp17,ac1,ac2,ac3, ac4, ac5, ac6,ac12, ac13, acep2, acep3, acep6, acep11, acep12, acep13, acep15, cal1, cal2, cal4, cal6, cal10, [37,38].

The exploratory structural equation model resulted in a good overall fit with 25 items. The fit indices indicate that the model is adequate: the RMSEA is within the desired range (RMSEA ≤ 0.08), while both the CFI and TLI exceed the recommended cutoff of >0.90, with values of 0.97 and 0.95, respectively. Additionally, the value of the SRMR is low (SRMR = 0.03), indicating an acceptable fit of the model to the data (Table 3).

**Table 3.** Summary of fit indicators from EERHC models.

| | $\chi^2$ | df | $p$-Value | RMSEA | CFI | TLI | SRMR |
|---|---|---|---|---|---|---|---|
| ESEM | 24,850 | 300 | 0.000 | 0.07 | 0.97 | 0.95 | 0.03 |
| CFI | 24,850 | 300 | 0.000 | 0.07 | 0.96 | 0.96 | 0.05 |

### 3.3. Confirmatory Factorial Analysis

Since the items were answered using a four-point ordinal scale, the confirmatory factorial analysis was carried out with the WLSMV estimator [33,37].

Model: The fit indices show that the initially proposed model is satisfactory ($\chi^2$ = 24,850, df = 300, $p$ = 0.000; RMSEA: 0.07; CFI = 0.96; TLI = 0.96; and SRMR = 0.046) (Table 3). Overall, the model of 25 items distributed across 4 factors, presents adequate fit indices. The reliability of the scale was estimated with Cronbach's α coefficient = 0.98 and McDonald's ω = 0.98 (Table 4).

**Table 4.** Factor loadings and correlations of the evaluated measurement models.

| Items | Element | ESEM F1 | ESEM F2 | ESEM F3 | ESEM F4 | CFA F1 | CFA F2 | CFA F3 | CFA F4 |
|---|---|---|---|---|---|---|---|---|---|
| Disp3 | My health care site has enough rooms for the number of people coming through | **0.37** | −0.01 | 0.18 | 0.19 | **0.69** | | | |
| Disp5 | My health care site has adequate hygiene in its bathrooms and other facilities | **0.57** | −0.11 | 0.35 | −0.03 | **0.73** | | | |
| Disp7 | My health care site where I receive care provides me with all the medications prescribed by the health team. | **0.69** | 0.03 | 0.13 | −0.05 | **0.76** | | | |
| Disp8 | My health care site where I receive care has the necessary equipment for performing the exams that have been recommended to me. | **0.91** | 0.07 | −0.05 | −0.11 | **0.8** | | | |
| Disp9 | My health care site has modern equipment for my care | **0.84** | 0.04 | −0.02 | 0.01 | **0.84** | | | |
| Disp10 | My health care site where I receive care schedules a reasonable appointment time for me to take the exams requested by the professional. | **0.87** | 0.01 | −0.15 | 0.01 | **0.73** | | | |
| Disp12 | My health care site has enough personnel to handle its patients | **0.58** | 0.08 | 0.01 | 0.15 | **0.78** | | | |

| Items | Element | ESEM | | | | CFA | | | |
|-------|---------|------|------|------|------|------|------|------|------|
| | | F1 | F2 | F3 | F4 | F1 | F2 | F3 | F4 |
| Disp16 | My health care site has personnel who give indications and referrals to ensure treatment continuity for me | **_0.5_** | 0.03 | 0.15 | 0.21 | **_0.83_** | | | |
| Ac7 | My health care site has prices which do not keep me from using services | −0.08 | **_0.74_** | 0.18 | 0.03 | | **_0.83_** | | |
| Ac8 | My health care site lets me access care with the professional I need, without being stopped by costs | −0.15 | **_0.95_** | 0.92 | 0.05 | | **_0.92_** | | |
| Ac9 | My health care site lets me access exams in the location without being impeded by costs | 0.1 | **_0.94_** | −0.08 | −0.01 | | **_0.95_** | | |
| Ac10 | My health care site lets me access treatment within the establishment without being blocked by costs | 0.09 | **_0.9_** | −0.01 | −0.04 | | **_0.93_** | | |
| Ac11 | My health care site lets me access needed health care, even without the necessary documentation | 0.12 | **_0.62_** | −0.03 | 0.14 | | **_0.82_** | | |
| Acep1 | In my health care site, the personnel use understandable words to provide information | 0.14 | −0.12 | **_0.41_** | 0.31 | | | **_0.69_** | |
| Acep4 | My health care site has signs (information) in my language | −0.09 | 0.07 | **_0.82_** | 0.01 | | | **_0.77_** | |
| Acep5 | My health care site has forms and documents which I can understand | 0.03 | 0.02 | **_0.89_** | −0.01 | | | **_0.89_** | |
| Acep7 | My health care site where I receive care provides support material (brochures) with understandable information | 0.08 | 0.09 | **_0.81_** | −0.05 | | | **_0.86_** | |
| Acep8 | In my health care site, if I do not understand something about what I must do, someone can tell me where to ask | 0.2 | 0.01 | **_0.61_** | 0.05 | | | **_0.82_** | |
| Acep9 | The health care facility where I receive care, the administrative team shows respect for my customs (diet, clothing, vocabulary, child-rearing practices) | 0.17 | 0.2 | **_0.45_** | 0.08 | | | **_0.85_** | |
| Cal3 | In my health care site facility where I receive care, the staff identifies themselves at the moment of my attention. | 0.08 | 0.12 | −0.07 | **_0.54_** | | | | **_0.65_** |
| Cal5 | In my health care site, the doctor/professional takes enough time to examine me and give directions | 0.1 | 0.09 | 0.01 | **_0.67_** | | | | **_0.83_** |
| Cal7 | In my health care site, the personnel explain the procedures to be done during my care and treatment | −0.16 | 0.03 | 0.15 | **_0.91_** | | | | **_0.9_** |
| Cal8 | In my health care site, the personnel clearly explain treatment options to me | −0.03 | −0.01 | −0.02 | **_0.97_** | | | | **_0.9_** |
| Cal9 | In my health care site, the personnel explain adverse side effects of the medications or procedures prescribed | 0.16 | −0.07 | −0.09 | **_0.82_** | | | | **_0.82_** |
| Cal11 | In my health care site, the personnel make me feel safe and confident during their procedures | 0.12 | 0.16 | 0.16 | **_0.48_** | | | | **_0.87_** |
| F1: | Availability | **1.00** | | | | **1.00** | | | |
| F2: | Accessibility | 0.66 | **1.00** | | | 0.72 | **1.00** | | |
| F3: | Acceptability | 0.64 | 0.59 | **1.00** | | 0.79 | 0.7 | **1.00** | |
| F4: | Quality | 0.71 | 0.63 | 0.6 | **1.00** | 0.78 | 0.71 | 0.73 | **1.00** |

## 4. Discussion

The objective of this study was to analyze the internal structure of the EERHC scale, based on the focus on the right to health care from the WHO, from the perspective of international migrants, in Chile, thereby providing preliminary empirical evidence about the validity of this scale.

The initial analysis of the 54-item model using exploratory structural equation modeling (ESEM) allowed us to identify low standardized factorial loads, leading to item purification and the final model. As a result, the scale of experience in exercising the right to health care (EERHC) consists of 25 items distributed across 4 factors. The final version of the instrument consisted of eight items for availability, five for accessibility, six for acceptability, and six for quality. The EERHC instrument reported satisfactory psychometric properties.

While there are currently instruments describing aspects of the health care service process, these are centered on measuring user satisfaction and health care quality and are useful when developing projects to improve health care centers' administrative aspects [11–13]. International studies have reported greater user satisfaction and have reported association with the geographical zone, the size of the health care center, and administrative aspects including providing information and the availability of treatment and/or possibility of performing exams [26]. In Latin America, some studies have used the SERVQUAL scale to measure service quality. This scale records five dimensions: tangibility (evaluating infrastructure, personnel presentation, cleanliness, number, and comfort of beds); reliability (identifying wait times, schedule fulfillment, and trust instilled); response capacity (employees' disposition to answer questions, procedural simplicity, inter-employee cooperation); safety (evaluating personal protection use); and finally, the empathy dimension, considering health team behavior [11,13,27].

While there are other instruments in the literature, the construct used in the measurement would be only one dimension of the rights focus. By contrast, this new instrument intends to contribute to the literature by providing measurements of tangible and intangible aspects perceived by international migrants who use health care services. This will permit measurement of elements rooted in health care practices such as standardized coverage and use of technical language, as well as ethical aspects of health care practices [28–30].

EERHC development responds to the need to have an instrument aimed at measuring the level of the right to health care which has been achieved by international migrants using primary health care centers. The relevance of this first care level, according to the WHO, is due to the contact it must achieve with the community as the gateway for the health system and primary patient contact [16,30,39]. Given the importance of this care level, it is relevant to monitor how the focus on the right to health care has been implemented, given the different dimensions and contexts involved.

The proposal of this study helps us know about more concrete aspects of the right to health care and how it has been implemented in various first-level centers. This includes the indicators relating to the process whose goal is to regulate the flow of people into other care levels via their resolution, containment, and derivation to different levels to respond to their health needs [16–40] as well as the ethical aspects arising in relations between personnel and international migrants [29,30].

The limitations of the study lie in the international migrant population responding to the survey, which mainly comprises Latin American people from countries such as Colombia and Venezuela. Another limitation is in the non-probabilistic sample type, which limits the findings' representativity. The results let us recognize the need to further investigate health care practices, in order to reduce the gap between theory and praxis.

## 5. Conclusions

The EERHC scale has twenty-five items and four dimensions, corresponding to availability accessibility, acceptability, and quality and safety based on the literature review, specifically though the goodness of fit indices, indicating that the factorial model is reliable, valid, and correctly fits the factors.

Our study results imply the inclusion of indicators often ignored in process and/or results measurement, while also allowing us to verify problems in implementing programs and/or regulations.

Future research is encouraged to use the EERHC to generate an evidence that enables measuring aspects of health care both the system and the human component by associating ethical elements.

**Author Contributions:** Conceptualization, C.C.-R. and A.U.; methodology, C.C.-R.; software, C.C.-R.; validation, C.C.-R., A.U. and C.L.; formal analysis, C.C.-R.; investigation, C.C.-R. and A.U.; resources, C.C.-R. and A.U.; data curation, C.C.-R. and A.U.; writing—original draft preparation, C.C.-R.; writing—review and editing, C.C.-R. and A.U.; visualization, C.C.-R. and C.L.; supervision, A.U.;

project administration, C.C.-R.; funding acquisition, C.C.-R. and A.U. All authors have read and agreed to the published version of the manuscript.

**Funding:** This research received no external funding.

**Institutional Review Board Statement:** The study was conducted in accordance with the Declaration of Helsinki and approved by the Institutional Review Board (or Ethics Committee) of Universidad Católica del Norte (protocol code 015/2021 and date 29 July 2021 of approval).

**Informed Consent Statement:** Informed consent was obtained from all subjects involved in the study.

**Data Availability Statement:** The data is not available.

**Conflicts of Interest:** The authors declare no conflict of interest.

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
