# Peer review of "Scale Measurement of Health Primary Service Utilization among the Migrant International Population"

_ejihpe, doi:10.3390/ejihpe13050064_

Round 1

Reviewer 1 Report

The paper sounds interesting and published timely compared to the field work done and for the importance in its proper field of study. However, Authors need to major improve the organization of contents: 

- Abstract: The acronym ERHC needs to be explicit from the abstract not at pag. 3 since it's the core topic of your paper.

- 1. Introduction: statistics on international migration need to be up-to-date. The article presents an argumentative style that is sometimes unscientific, with often apodictic statements or not empirically supported.

- 4. Discussion: the paper does not take in consideration the proper scholarship on the relation between migration and health, in particular neither in the Introduction nor in the Discussion, the Results have been discussed in a comparative perspective. I suggest Authors to deepen the comparative perspective of similar studies in Latin America, reporting the main results in order to make understand to readers the relevance of their results.

- 5. Conclusion: they seem trivial and not developed.

Author Response

Estimado revisor,

"Consulte el archivo adjunto".

Reviewer 2 Report

Dear Authors,

First of all, congratulations for elaborating this work on migration and health.

The abstract in English need to be improved. There are sentences without subject.

In the second paragraph authors stated that health improves or declines, this needs further argumentation. You might mention social determinants of health, also considering that you are quoting WHO, and access to healthcare is one of these determinants. Then, authors citation 5 from line 34-36 needs further development, it is not clear the thesis that the authors want to state.

 The four dimensions from line 45-58 need to be reformulated. Also some literature regarding the importance of each dimension for the delivery of healthcare for migrant population might help to illustrate the soundness of each of them for this population.

In the methodology, you use the word “availability” to explain “accessibility” this overlap between dimensions might complicate the operationalization of concepts.

Examples of the questions to illustrate each dimension would be welcome. I guess the tool is used to measure the experience of the patient in the whole healthcare system (this needs to be clarified, and also be written as a limitation but also a potential future study, the adaptation for services or care levels) .

In the discussions, authors claim “to have a validate instrument to objectively measure the right to health care among international migrants facing the first level of care, which is the main gateway to the national health system”, there is not a way to measure the right to health care, this would be “the degree of fulfilment the right of healthcare”. But then, you such present to the reader how to interpret the results of your scale. As you might know, professionals and academics need practical tools.

Author Response

Dear Reviewer,

Reviewer 3 Report

This study analyzed the internal structure of the ERHC scale, based on the focus from the WHO of the right to health care and from the perspective of international migrants in Chile, which showed an interesting topic for the practice. However, the manuscript needs some substantial revisions in my opinion.

1. Please revise the abstract. The presentation of abstracts should refer to the specific requirements of the journal.

2. Please give the full text of the abbreviation when it first appears in the body of the paper.

3. Please distinguish between numbers appearing in the text (for example, lines 107-126) and the number of the title.

4. Section 2.1 is too brief. Although this paper argued that the study is an instrumental study, the authors should also provide a clear research framework. The research process can be illustrated in combination with the contents of Section 2.2-2.6.

5. The variable selection process under each dimension needs to be supplemented.

6. The quantities of the variables in Table 2 seem superfluous. The authors should explain how these variables perform and discuss whether the variables need to be filtered. If there is a filtering, the results should be presented. Correspondingly, the variables in Figure 1 are still too overlapped to identify the variables.

7. Please further explain the meaning of the results in Table 3.

8. How did the authors construct Model? The modeling process is not explicitly described.

9. The WLSMV method is mentioned in the article, but there seem no results related in the article.

10. The conclusion is too short. According to the statement, this study is more like a reproduction of the existing research methods, and it is also difficult to see the originality of the paper.

Author Response

Dear Reviewer,

Round 2

Reviewer 1 Report

The comments have been addressed.

Author Response

Dear,

I am writing to express my sincere gratitude for your valuable work as a reviewer. I deeply appreciate the time and effort you dedicated to the review of my article. Your comments and suggestions were extremely helpful and allowed me to significantly improve my work.

Again, thank you very much for your work and commitment.

Reviewer 2 Report

Congrats

Author Response

(The authors gave the same response as above.)

Reviewer 3 Report

This paper has been improved to some extent. After a full text check again, this paper can be considered for publication.

Author Response

Dear,

I am writing to express my sincere gratitude for your valuable work as a reviewer. I deeply appreciate the time and effort you dedicated to the review of my article. Your comments and suggestions were extremely helpful and allowed me to significantly improve my work.

Again, thank you very much for your work and commitment.

"Please see the attachment".
